# Understanding when Dynamics-Invariant Data Augmentations Benefit Model-Free Reinforcement Learning Updates

**Nicholas E. Corrado**
Department of Computer Sciences
University of Wisconsin – Madison
`ncorrado@wisc.edu`

**Josiah P. Hanna**
Department of Computer Sciences
University of Wisconsin – Madison
`jphanna@cs.wisc.edu`

## Abstract

Recently, data augmentation (DA) has emerged as a method for leveraging domain knowledge to inexpensively generate additional data in reinforcement learning (RL) tasks, often yielding substantial improvements in data efficiency. While prior work has demonstrated the utility of incorporating augmented data directly into model-free RL updates, it is not well-understood when a particular DA strategy will improve data efficiency. In this paper, we seek to identify general aspects of DA responsible for observed learning improvements. Our study focuses on sparse-reward tasks with dynamics-invariant data augmentation functions, serving as an initial step towards a more general understanding of DA and its integration into RL training. Experimentally, we isolate three relevant aspects of DA: state-action coverage, reward density, and the number of augmented transitions generated per update (the augmented replay ratio). From our experiments, we draw two conclusions: (1) increasing state-action coverage often has a much greater impact on data efficiency than increasing reward density, and (2) decreasing the augmented replay ratio substantially improves data efficiency. In fact, certain tasks in our empirical study are solvable only when the replay ratio is sufficiently low.

## 1 Introduction

Reinforcement learning (RL) algorithms are often data inefficient and often produce policies that fail to generalize outside of a narrow state distribution. Recently, a number of RL algorithms and applications have been published that leverage *data augmentation* to enhance convergence and generalization (Mitrano and Berenson, 2022; Pitis et al., 2020; Qiao et al., 2021). Data augmentation (DA) is a technique in which agents generate additional synthetic experience by applying transformations to their observed experience. Since augmented data can be generated without the expense of additional interaction with the environment, it is an attractive technique for improving the *data efficiency* of RL algorithms (*i.e.*, the number of environment interactions needed to solve a task).

Much of the prior work in DA for RL (Hansen and Wang, 2021; Laskin et al., 2020; Raileanu et al., 2021; Wang et al., 2020; Yarats et al., 2020; 2021) builds off of DA techniques used in computer vision (Chen et al., 2020). Other works have used domain-dependent DA strategies for non-visual tasks (Abdolhosseini et al., 2019; Mikhail Pavlov and Plis, 2018), including DeepMind's AlphaTensor (Fawzi et al., 2022) which uses RL to discover more efficient matrix multiplications. These works introduce methods for generating augmented data and frameworks for integrating it into RL that demonstrably improve training performance. To the best of our knowledge, most prior work on DA has focused on introducing new types of data augmentation functions and demonstrating that they can boost the data efficiency of RL. What is missing from the literature is a clear understanding of which aspects of DA yield improvements. Rather than adding to existing work by introducing new DA strategies, our main contribution is an investigation into the following question:

*When and why does data augmentation improve data efficiency in reinforcement learning?*

As a motivating example, consider using an off-policy RL algorithm to solve a 2D navigation task in which an agent must reach a random goal position (Fig. 1a). In this task, transitions observed by the agent can be augmented through either random translations of the agent (Fig. 1b) or random rotations of the agent and goal (Fig. 1c). As shown in Fig. 1d, if we double the agent's learning data via DA and double the batch size used for updates, we achieve significant improvements in data efficiency compared to learning without DA. Furthermore, agents that learn from extra augmented data even surpass the performance of agents that learn from an equal amount of extra real data collected through additional environment interactions. More concretely, we double the amount of policy data collected between updates and double the batch size used for updates so that non-augmented agents learn from the same amount of data and perform the same number of updates as the augmented agents. As shown in Fig. 1d, additional augmented data leads to faster learning than simply collecting an equal amount of additional data from the agent's policy. In fact, doubling the learning data via the translation augmentation is nearly as good as learning from 8 times as much policy-generated data. Thus, these augmentations must offer benefits beyond what additional policy-generated data can offer. An understanding of which aspects of DA yield these benefits will serve as an initial step towards guiding practitioners on how to more effectively incorporate DA into RL.

DA has taken many forms in the RL literature (Andrychowicz et al., 2017; Hansen and Wang, 2021; Laskin et al., 2020; Pitis et al., 2020; Qiao et al., 2021), and a comprehensive analysis of different DA frameworks, tasks, and data augmentation functions is beyond the scope of a single study. Thus, in this work, we instead aim to better understand the benefits of integrating dynamics-invariant augmented data directly into model-free off-policy RL updates. With this focus in mind, we must leave studies on DA frameworks with auxiliary tasks (Hansen and Wang, 2021; Hansen et al., 2021; Raileanu et al., 2021; Wang et al., 2020), and studies on data augmentation functions that generate unrealistic data – such as visual data augmentations (Laskin et al., 2020) – for future work.

Our investigation focuses on three aspects of DA that we hypothesize influence learning in sparse-reward tasks: an increase in state-action coverage via DA, the amount of additional reward signal generated via DA (reward density), and the number of augmented transitions generated per update (the augmented replay ratio). State-action coverage and reward density relate to how DA affects the agent's distribution of learning data, whereas the augmented replay ratio relates to how augmented data is incorporated into RL training. We empirically ablate the effects of these factors using a simple and controllable DA framework similar to frameworks found in existing work (Laskin et al., 2020; Pitis et al., 2020).[1] In summary, our contributions are:

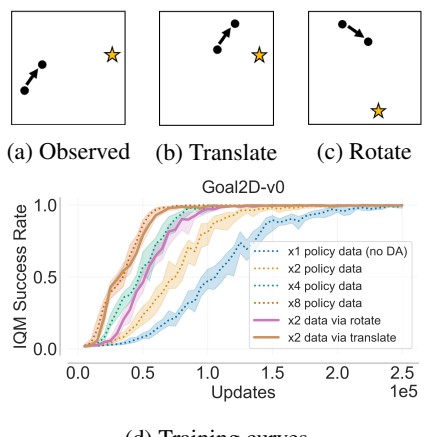

(a) Observed    (b) Translate    (c) Rotate

(d) Training curves

Figure 1: Visualizations of two augmentations – translation (1b) and rotation (1c) – for a 2D navigation task in which an agent (black dot) must reach a goal (gold star). In 1d, "x$N$ policy data" corresponds to collecting $N$ times as many transitions with the agent's current policy between updates, and "x2 via rotate/translate" corresponds to generating one augmented transition per observed transition. We increase the batch size and replay buffer sizes proportionally to the amount of extra data to keep the replay ratio and replay age fixed across all experiments. We plot the interquartile mean success rate over 50 seeds with 95% bootstrap confidence belts.

1. We introduce a framework for studying DA in RL that is amenable to analysis.

2. While it is widely understood that high state-action coverage and discovery of reward signal are critical to data efficient RL, our experiments show that increasing state-action coverage via DA often has a much greater impact on data efficiency than increasing reward density.

3. We show that the success of DA depends strongly on the augmented replay ratio. In fact, certain tasks in our empirical study are solvable only when the augmented replay ratio is sufficiently low.

---

[1]Code available at https://github.com/Badger-RL/UnderstandingDataAugmentationForRL

## 2 RELATED WORK

In this section, we provide an overview of data augmentation techniques and applications in RL.

**Dynamics-based Augmentation:** Several prior works use data augmentation functions that affect the agent's current state, action, and next state. Pitis et al. (2020; 2022) stitch together locally independent features of different transitions to generate additional data and provides a method for identifying local independence. Many model-based algorithms learn from synthetic data generated by a learned dynamics model and can be viewed as DA methods (Gu et al., 2016; Racanière et al., 2017; Sutton, 1990; Venkatraman et al., 2016).

**Hindsight Experience Replay:** In goal-conditioned RL, Hindsight Experience Replay (HER) (Andrychowicz et al., 2017; Fang et al., 2018; Liu et al., 2019; Rauber et al., 2017) counterfactually relabels the goal of a trajectory to generate additional data. This technique can be applied when transition dynamics are independent of the agent's goal, as is often the case. Follow-up work on HER has demonstrated that *hindsight bias* caused by changing the distribution of observed goals many hinder learning (Lanka and Wu, 2018; Li et al., 2020).

**Applications of Domain-Specific Data Augmentation:** Several recent works have leveraged domain-knowledge to create new data augmentation functions. Mikhail Pavlov and Plis (2018) and Abdolhosseini et al. (2019) apply DA to locomotion problems in which an optimal policy has a symmetric gait, and Mitrano and Berenson (2022) focus on augmenting trajectories of poses and movable objects relevant in robot manipulation. Qiao et al. (2021) consider DA in the context of differentiable simulation to generate additional approximately correct transitions (a method they refer to as sample enhancement). DeepMind's AlphaTensor (Fawzi et al., 2022) exploits two invariances: tensor decompositions are commutative, and tensor rank is invariant to the ordering of rows and columns. They exploit commutativity by generating additional augmented transitions and rank invariance using a network that disregards the row and column ordering of input tensors.

**State-based Augmentation:** Much of the prior work in DA for RL focuses on augmenting visual observations (Guan et al., 2021; Wang et al., 2020; Yarats et al., 2021). Laskin et al. (2020) train RL agents on multiple views of visual states (crops, recolorations, rotations, etc.). Raileanu et al. (2021) introduce regularizers to ensure an agent's policy and value function are both invariant under augmentation. Sinha et al. (2022) ensure small perturbations of non-visual states state have similar state-action values. Hansen and Wang (2021) learn a state representation that is invariant under augmentation rather than directly using the augmented data for policy optimization. Hansen et al. (2021) identify sources of instability when performing visual DA. This line of work relates to domain randomization (Peng et al., 2018; Tobin et al., 2017), as agents are trained to be robust to randomized augmentations of observations. Visual augmentations are beyond the scope of our study; we focus on integrating augmented data that respects the environment's dynamics into model-free updates, but visual augmentations generate unrealistic data and are typically used for auxiliary representation learning tasks. We provide further discussion on visual augmentations in Appendix A.

**Invariant Model Architectures:** DA often – though not always – exploits known invariances within the environment's state space and/or dynamics. In this case, an alternative to DA is to simply hard-code these invariances into the agent's policy model (van der Pol et al., 2020; 2021; Wang et al., 2022). Residual Pathway Priors (RPPs) (Finzi et al., 2021) capture invariances using a soft prior, biasing agents toward invariant policies without constraining them.

While these prior works focus on developing data augmentation functions or methods for incorporating augmented data into RL training, our work introduces a framework to investigate when and why DA improves learning.

## 3 PRELIMINARIES

In this section, we formalize the RL setting and the class of data augmentation functions we use.

### 3.1 REINFORCEMENT LEARNING

We consider finite horizon Markov decision processes (MDPs) (Puterman, 2014) defined by $(\mathcal{S}, \mathcal{A}, p, r, d_0, \gamma)$ where $\mathcal{S}$ and $\mathcal{A}$ denote the state and action space, respectively, $p(\boldsymbol{s}' \mid \boldsymbol{s}, \boldsymbol{a})$ denotes

the probability density of the next state $s'$ after taking action $a$ in state $s$, and $r(s, a)$ denotes the reward for taking action $a$ in state $s$. We write $d_0$ as the initial state distribution, $\gamma \in [0, 1)$ as the discount factor, and $H$ the length of an episode. We consider stochastic policies $\pi_\theta : \mathcal{S} \times \mathcal{A} \to [0, 1]$ parameterized by $\theta$. The RL objective is to find a policy that maximizes the expected sum of discounted rewards $J(\theta) = \mathbb{E}_{\pi_\theta, s_0 \sim d_0} \left[ \sum_{t=0}^{H} \gamma^t r(s_t, a_t) \right]$.

## 3.2 DATA AUGMENTATION FUNCTIONS

In the literature, data augmentation functions (DAFs) have taken different forms and served different purposes. We introduce a few important definitions to help classify the DAFs we focus on.

**Definition 1.** A transition $(s, a, r, s')$ is *valid* if it is possible under the transition dynamics and reward function, *i.e.* $p(s' \mid s, a) > 0$, and $r = r(s, a)$.

**Definition 2.** Let $\mathcal{T} \subset \mathcal{S} \times \mathcal{A} \times \mathbb{R} \times \mathcal{S}$ denote the set of possible transitions and let $\Delta(\mathcal{T})$ denote the set of distributions over $\mathcal{T}$. A *data augmentation function* (DAF) is a stochastic function $f : \mathcal{T} \to \Delta(\mathcal{T})$ mapping a transition $(s, a, r, s')$ to an augmented transition $(\tilde{s}, \tilde{a}, \tilde{r}, \tilde{s}')$.

**Definition 3.** A DAF is *dynamics-invariant* if it is closed under valid transitions.

We focus on dynamics-invariant DAFs so that augmented data agrees with the underlying MDP, since learning from data that does not match the MDP's dynamics can harm learning (Moerland et al., 2023). This focus includes methods such as HER (Andrychowicz et al., 2017) and CoDA (Pitis et al., 2020) which provide domain-independent DAFs. However, we do not restrict ourselves to domain-independent DAFs as, in practice, domain-experts may be able to produce dynamics-invariant DAFs even though they cannot identify an optimal domain policy (Abdolhosseini et al., 2019; Fawzi et al., 2022; Mikhail Pavlov and Plis, 2018). Our focus does exclude some recent works on DA – especially those focusing on visual augmentations (Laskin et al., 2020; Raileanu et al., 2021) – that produce augmented states that would never be observed in simulation such that $p(\tilde{s}'|\tilde{s}, \tilde{a}) = 0$, since these augmentations do not satisfy our definition of valid. We note that dynamics-invariant DAFs will not necessarily preserve transition probabilities in tasks with stochastic dynamics. We elaborate on the widespread applicability of dynamics-invariant DAFs in Appendix A.

## 4 A FRAMEWORK FOR STUDYING DATA AUGMENTATION IN RL

---

**Algorithm 1** Off-Policy RL with Data Augmentation

---

**Inputs:** Data augmentation function $f$, augmentation ratio $m$, update ratio $\alpha$, batch size $b$
Initialize policy $\pi_\theta$, replay buffer $\mathcal{R}$, and augmented replay buffer $\widetilde{\mathcal{R}}$
**for** $t = 1, 2, \ldots$ **do**
    Collect transition $(s_t, a_t, r_t, s_{t+1})$ using $\pi_\theta$
    Append $(s_t, a_t, r_t, s_{t+1})$ to $\mathcal{R}$
    **for** $i = 1, \ldots, m$ **do**
        $(\tilde{s}_t, \tilde{a}_t, \tilde{r}, \tilde{s}_{t+1}) \sim f(s_t, a_t, r_t, s_{t+1})$
        Append $f(\tilde{s}_t, \tilde{a}_t, \tilde{r}_t, \tilde{s}_{t+1})$ to $\widetilde{\mathcal{R}}$
    **if** update **then**
        Sample mini-batch $\mathcal{D}$ of $b$ transitions from $\mathcal{R}$
        Sample mini-batch $\widetilde{\mathcal{D}}$ of $b\alpha$ transitions from $\widetilde{\mathcal{R}}$
        Update policy and/or value function using $\mathcal{D} \cup \widetilde{\mathcal{D}}$

---

Prior work on DA in RL not only considers different DAFs but also various methods for integrating the augmented data into RL algorithms. To focus our study, we introduce a specific framework (Algorithm 1) for incorporating augmented data into the training loop of any off-policy RL algorithm. An off-policy algorithm is essential, as augmented data may not be distributed according to the state-action distribution of the current policy. We focus on the effects integrating augmented data directly into policy and/or value function updates.

Within our framework, the agent observes a transition, applies a given DAF $f$ to that transition to generate some number of augmented transitions, and then stores the observed and augmented transitions in separate replay buffers – the observed and augmented replay buffers, respectively. When performing policy and/or value function updates, the agent samples data from both buffers and combines the data for the updates. To control how much augmented data we generate and use in each update, we introduce a few parameters into the framework.

The *augmentation ratio*, $m$, specifies the number of augmented transitions generated per observed transition. Some DAFs, such as the translation augmentation in Fig. 1b, can produce multiple unique augmentations from the same input transition. Each time the agent observes one real transition,

$m$ augmented transitions are sampled from the DAF. We use this parameter to study whether it is beneficial to produce multiple augmentations to diversify the augmented replay buffer. When the augmentation ratio is increased, we increase the augmented replay buffer size proportionally such that the age of the oldest observed and augmented transitions remain equal.[2]

The *update ratio*, $\alpha$, denotes the the ratio of augmented to observed data used for updates, *e.g.*, $\alpha = 1$ denotes that half of the data used for each update is augmented data. With access to large amounts of augmented data, it may be beneficial to increase the amount of augmented data used in updates. However, a large update ratio may exacerbate the *tandem effect* (Ostrovski et al., 2021), a decrease in performance when learning predominantly from data not collected by the agent.

The augmentation ratio modulates a third relevant quantity: the number of updates per augmented transition generated, or *augmented replay ratio*.[3] In the absence of DA, Fedus et al. (2020) found that found that decreasing the replay ratio of observed data (the *observed replay ratio*) can improve data efficiency, though other works have improved data efficiency by developing techniques that enable learning with large replay ratios (Chen et al., 2021; D'Oro et al., 2023; Nikishin et al., 2022; Scheller et al., 2020). One can decrease the augmented replay ratio by increasing the augmentation ratio ($m$) while keeping the frequency of policy and/or value function updates fixed.

Though a variety of methods exist for incorporating augmented data into RL, our framework offers several core benefits for our study:

1. **Control:** One can easily control the replay ratio and update ratio. Having control over the update ratio is especially important to ensure the agent uses a sufficient amount of observed data in each update to mitigate the tandem effect. Moreover, since augmented data is stored in a replay buffer and not sampled online, it is possible to keep the replay ratio of the augmented data equal to that of the observed data.

2. **Lower Systematic Variance:** Each update uses the same ratio of augmented to observed data (*i.e.* update ratio), and the same number of augmented transitions are generated for every observed transition, eliminating a possible source of training variation.

3. **Computational Efficiency:** By storing and reusing augmented data, we improve computational efficiency. While some DAFs can produce multiple unique augmentations of the same input transition, many can only produce a single augmentation – such as a reflection – in which case it is more efficient to reuse augmented data rather than generate new samples online every update.

Our framework is similar to those used in CoDA (Pitis et al., 2020), RAD (Laskin et al., 2020), and HER (Andrychowicz et al., 2017), as all three incorporate augmented data directly into updates without auxiliary tasks. We note that RAD as well as popular implementations (Raffin et al., 2021; Weng et al., 2022) of HER generate augmented samples during updates and discard them after use, whereas we save augmented data for reuse. Since we focus on using augmented data for model-free updates, Algorithm 1 is not intended to capture methods that use augmented data for auxiliary tasks but easily extends to include such methods (Hansen and Wang, 2021; Raileanu et al., 2021).

## 5 DISENTANGLING PROPERTIES OF DATA AUGMENTATION

We identify aspects of DA that we hypothesize may impact its effectiveness within our framework.

**State-Action Coverage:** DAFs can generate data that the current policy otherwise might not observe, increasing state-action coverage. Greater state-action coverage via DA may aid exploration. However, it may also generate data that is very off-policy with respect to the current policy and hence increase the variance of learning due to the tandem effect (Ostrovski et al., 2021).

---

[2]The age of a transition is the number of gradient steps taken by the agent since that transition was generated (Fedus et al., 2020).

[3]Prior works (D'Oro et al., 2023; Fedus et al., 2020; Nikishin et al., 2022; Scheller et al., 2020) define the replay ratio as the number of updates per environment interaction, characterizing how much the agent learns from existing data versus new experience. However, augmented data is generated by a DAF and can produce multiple augmentations per observed transition. Thus, for our analysis, the number of updates per augmented transition generated is a more appropriate metric.

**Reward Density:** Long horizon, sparse reward tasks are notoriously difficult since an RL agent is unlikely to discover reward signal through random exploration. A DAF that can produce transitions with additional reward signal could improve data efficiency. However, it is also known that reward-generating DA strategies such as HER (Andrychowicz et al., 2017) can bias learning and lead to overestimation of state-action values (Lanka and Wu, 2018) . For sparse reward tasks, we define *reward density* as the fraction of transition data in both observed and augmented replay buffers which successfully solve the task and thus contain reward signal.[4]

**Augmented Replay Ratio:** Some DAFs (such as the translation DAF in Fig. 1b) can generate multiple augmented transitions given a single input transition, substantially increasing the amount of data available to the agent. We hypothesize that it may be beneficial to generate as many augmented transitions as possible to lower the augmented replay ratio (Fedus et al., 2020).

While it is widely understood that high coverage and discovery of reward signal are critical to solving sparse-reward RL tasks (Andrychowicz et al., 2017; Tang et al., 2017), the degree to which increasing state-action coverage and reward density via DA *individually* contribute to data efficient RL is less clear. These factors are be difficult to completely isolate; since the reward function $r(\boldsymbol{s}, \boldsymbol{a})$ depends on $\boldsymbol{s}$ and $\boldsymbol{a}$, altering reward density necessarily changes state-action coverage. In our experiments, we attempt to isolate all three aspects of DA to determine how much each affects data efficiency.

## 5.1 EXPERIMENTS

We focus our experiments on four sparse-reward, continuous action `panda-gym` tasks (Gallouédec et al., 2021): **PandaPush-v3**, **PandaSlide-v3**, **PandaPickAndPlace-v3**, and **PandaFlip-v3** (Fig. 2), which we henceforth refer to as the Push, Slide, PickAndPlace, and Flip tasks, respectively. We consider two DAFs:

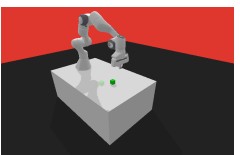

Figure 2: PandaPush-v3. A robotic arm must push a block to a goal location.

1. TRANSLATEGOAL: Relabel the goal with a new goal sampled uniformly at random from the goal distribution.

2. TRANSLATEGOALPROXIMAL($p$): Relabel the goal with a new goal sampled from the goal distribution. With probability $p$, the new goal is sufficiently close to the object to generate reward signal.

We also consider a toy 2D navigation task **Goal2D-v0** (Fig. 1a) in which an agent must reach a random goal. The agent receives reward $+1$ when it is sufficiently close to the goal and reward $-0.1$ otherwise. Agent and goal positions are initialized uniformly at random. We consider three DAFs:

1. ROTATE: Rotate the agent and goal by $\theta \in \{\frac{\pi}{2}, \pi, \frac{3\pi}{2}\}$.

2. TRANSLATE: Translate the agent to a random position.

3. TRANSLATEPROXIMAL($p$): Translate the agent to a random position. With probability $p$, the agent's new position is sufficiently close to the goal to generate reward signal.

These DAFs offer an avenue to investigate the role of reward density and state-action coverage in DA. For instance, one can modify reward density through $p$ in TRANSLATEGOALPROXIMAL($p$). Moreover, these DAFs are extremely general can be applied to many tasks, *e.g.* most navigation tasks. We include full descriptions of each environment and DAF in Appendices B and C, respectively. We use DDPG (Lillicrap et al., 2015) for Panda tasks and TD3 (Fujimoto et al., 2018) for Goal2D. Further training details are in Appendix G. We include experiments studying how all three factors affect an agent's generalization ability in Appendix E and include experiments studying the augmented replay ratio for dense reward MuJoCo tasks (Brockman et al., 2016) in Appendix F.

### 5.1.1 BENCHMARKING DATA AUGMENTATION

We first benchmark the performance of DA against simply collecting more policy data to establish how much our chosen DAFs improve data efficiency. Prior work has demonstrated that learning with augmented data is often more data efficient than learning without it, though it is unclear how learning

---

[4]With dense rewards, one may need to consider the full distribution of rewards in the replay buffer instead.

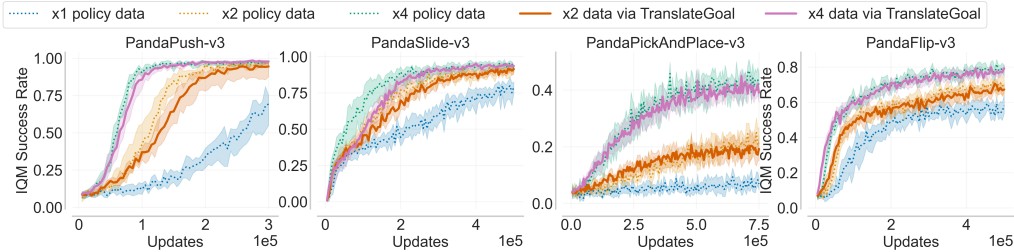

Figure 3: "x2 policy data" agents double their learning data by collecting twice as many samples between updates, and "x2 data via TranslateGoal" agents double their learning data by generating one augmented transition per observed transition. Note that the horizontal axis shows the number of updates rather than timesteps. Each curve shows the interquartile mean over 10 seeds. Shaded regions denote 95% bootstrap confidence belts.

from augmented data compares to learning from additional policy-generated data, as policy-generated data and augmented data are distributed differently in general. In these experiments, we increase the available learning data using DA or by collecting more data with the agent's current policy between updates. We label agents according to how many additional environment interactions they perform, *e.g.* "x2 policy data" corresponds to one extra environment interaction, and "x2 data via TRANSLATEGOAL" corresponds to generating one augmented transition per observed transition ($m = 1$). When collecting or generating extra data, we increase the batch size and replay buffer size proportionally so that all agents learn with the same amount of data, the same observed and augmented replay ratios, and the same replay age. Thus, augmented and observed data are treated equally in training.

From Fig. 3, we see that TRANSLATEGOAL offers significant improvement over no additional data. For instance, in Push, doubling the learning data via TRANSLATEGOAL doubles data efficiency. Though, additional policy-generated data generally yields equal or better performance. Having established that TRANSLATEGOAL improves data efficiency, in the following sections, we study the degree to which reward density and state-action coverage are responsible for these improvements.

### 5.1.2 STATE-ACTION COVERAGE

In this section, we study how increasing state-action coverage via DA affects data efficiency. Since the reward is a function of the agent's state and action, it is difficult to completely isolate the effects of increased coverage; a change in coverage affects reward density. Nevertheless, we can better understand the effect of increasing coverage by comparing agents trained using TRANSLATE and TRANSLATE-GOAL with agents trained using TRANSLATEPROXIMAL(0) and TRANSLATEGOALPROXIMAL(0) – DAFs that increase state-action coverage without generating additional reward signal. Early in training when there is little to no reward signal in the observed replay buffer, TRANSLATEGOAL(0) and TRANSLATEGOALPROXIMAL(0) have little affect on reward density. As the policy learns and more reward signal is added to the observed replay buffer, the lack of reward signal in the augmented replay buffer reduces the overall reward density. Thus, we can attribute any performance boost provided by TRANSLATEPROXIMAL(0) and TRANSLATEGOALPROXIMAL(0) to an increase in state-action coverage and/or a decrease in reward density. To better separate the effects of increased coverage and decreased reward density, we double the agent's training data using different ratios of augmented data to observed data (1:5, 1:2, and 1:1). A smaller split of TRANSLATEPROX-IMAL(0)/TRANSLATEGOALPROXIMAL(0) data corresponds to less coverage but also a smaller decrease in reward density. We report results for ratios yielding the largest improvements to data efficiency (*i.e.* ratios that best balances the increase in coverage with the decrease in reward density).

As shown in Fig. 4, TRANSLATEGOALPROXIMAL(0) in Slide, PickAndPlace, and Flip is as data efficient as TRANSLATEGOAL; increased coverage alone explains the benefits of TRANSLATEGOAL in these tasks. In Goal2D and Push, TRANSLATEPROXIMAL(0) and TRANSLATEGOALPROXIMAL(0) are more data efficient than no DA but less data efficient than TRANSLATE and TRANSLATEGOAL. Thus, although increased state-action coverage is the primary benefit in most tasks, we see that increased reward density can also play a role. In the following section, we further disentangle state-action coverage and reward density to assess how critical high reward density is for these tasks.

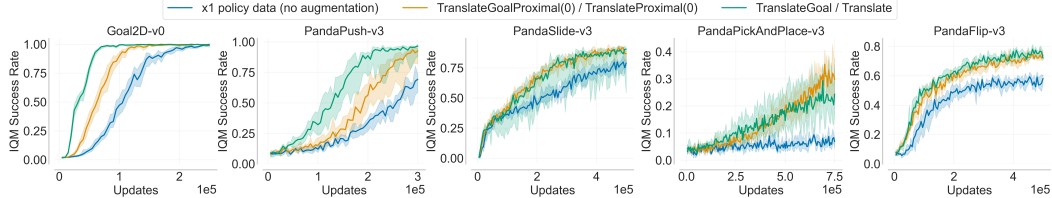

Figure 4: Learning with TRANSLATEGOAL, TRANSLATEGOALPROXIMAL(0), TRANSLATE, and TRANSLATEPROXIMAL(0). We plot the IQM success rate over 10 seeds for Panda experiments and 50 for Goal2D. Shaded regions denote 95% bootstrap confidence belts.

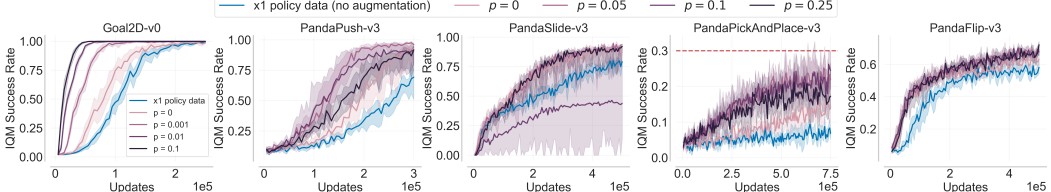

Figure 5: Learning with TRANSLATEGOALPROXIMAL($p$) for various $p$. Darker colors correspond to larger $p$ values. The dashed red line for PickAndPlace denotes the final IQM success rate achieved by TRANSLATEGOALPROXIMAL(0) in Fig. 4. We plot the IQM success rate over 10 seeds for Panda experiments and 50 for Goal2D. Shaded regions denote 95% bootstrap confidence belts.

### 5.1.3 REWARD DENSITY

We now strive to further disentangle state-action coverage from reward density by studying how changes to reward density affect learning. In the following experiment, we modify reward density by varying the probability $p$ of generating reward signal with TRANSLATEPROXIMAL($p$) and TRANSLATEGOALPROXIMAL($p$) while keeping the update ratio and augmentation ratio fixed at $\alpha = 1$ and $m = 1$, respectively. This setup does not completely isolate reward density from state-action coverage; changing reward density (increasing $p$) also affects state-action coverage, as it increases the amount of learning data in which the goal is near the object. Nevertheless, this experiment enables us to answer the following question: how critical is it that DA generates data with high reward density?

As shown in Fig. 5, changing $p$ has little effect on data efficiency in Slide and Flip, further supporting that increased coverage is the primary benefit of TRANSLATEGOAL in these tasks. In Push and PickAndPlace, $p = 0.05$ is most data efficient[5], while in Goal2D, data efficiency increases significantly as with $p = 0.01$, and increasing to $p = 0.1$ offers marginal additional improvement. In these three tasks, the largest $p$ values decrease data efficiency, since changes to the distribution of reward signal and can bias updates (*i.e.* hindsight bias (Lanka and Wu, 2018; Li et al., 2020)). Since the most data efficient learning occurs when DA contributes no reward signal or a relatively small amount, we conclude that high reward density is not critical to successful DA.

Collectively, our state-action coverage and reward density experiments suggest that *increasing state-action coverage via DA often has a much greater impact on data efficiency than increasing reward density.* Thus, our results suggest that RL practitioners choosing among candidate DAFs or designing new DAFs should focus on increasing state-action coverage more so than increasing reward density.

### 5.1.4 AUGMENTED REPLAY RATIO

Our previous experiments study how two properties of DAFs affect RL, though performance may be sensitive to *how* augmented data is incorporated into RL training. In this section, we study how the number of updates per augmented transition generated affects data efficiency.

Existing DA strategies often incorporate multiple augmentations of the same observed transition into policy optimization. For instance, HER (Andrychowicz et al., 2017) generates 4 hindsight transitions

---

[5]In Fig. 4, TRANSLATEPROXIMALGOAL(0) achieves an IQM success rate of 0.3 (red dashed line), outperforming all PickAndPlace agents in Fig. 5; high reward density augmented data is not critical in this task.

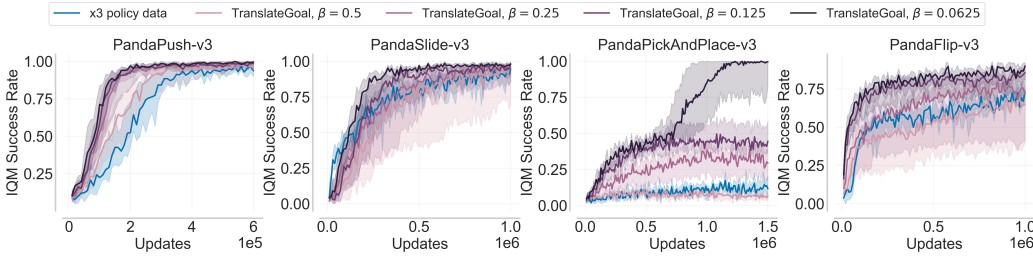

Figure 6: Decreasing the augmented replay ratio with the update ratio fixed at $\alpha = 2$ markedly improves learning. Darker colors correspond to smaller replay ratios. We plot the IQM success rate over 10 seeds. Shaded regions denote 95% bootstrap confidence belts.

per observed transition, and CoDA (Pitis et al., 2020) generates up to 16 augmentations per observed transition. Generating additional augmentations decreases the *augmented replay ratio*, the number of updates per augmented transition generated. We hypothesize that decreasing the augmented replay ratio can provide a similar benefit to decreasing the replay ratio of observed data noted by Fedus et al. (2020). Decreasing the replay ratio of *observed* data can be expensive – requiring more environment interactions between updates – while decreasing the augmented replay ratio is comparatively cheap.

In this experiment, we decrease the augmented replay ratio $\beta$ by generating more augmentations per observed transition (*i.e.*, increasing $m$). We scale the augmented replay buffer size proportionally and keep the ratio of augmented to observed data used in updates fixed at $\alpha = 2$. As shown in Fig. 6, a lower augmented replay ratio alone substantially improves data efficiency and overall performance across all panda tasks. Moreover, a low replay ratio is *necessary* to solve PickAndPlace within our training budget; 100% success rate can be achieved with $\beta = 0.0625$, while no learning occurs with $\beta = 0.5$. A lower replay ratio also increases data efficiency in Goal2D with both TRANSLATE and ROTATE DAFs; due to space constraints, we include these figures in Appendix D.3. Decreasing the replay ratio is a preferable alternative to increasing the amount of augmented data used in updates, as the latter may exacerbate the tandem effect (Ostrovski et al., 2021) in which RL from passive data fails. We support this claim with additional experiments provided in Appendix D.1.

Empirically, we have demonstrated that a DAF's success may depend strongly on how we integrate its augmented data into training. To effectively apply DA, one must understand desirable properties of DAFs *and* relevant implementation details.

## 6 CONCLUSIONS, LIMITATIONS, AND FUTURE WORK

While prior work has demonstrated that incorporating augmented data in model-free off-policy reinforcement learning (RL) updates can improve the data efficiency of RL algorithms, we lack a clear understanding of which aspects of data augmentation (DA) yield such improvements. In this paper, we isolated three aspects of DA in sparse reward tasks with dynamics-invariant data augmentation functions (DAFs) – state-action coverage, reward density, and the replay ratio of augmented data (the number of augmented samples generated per timestep) – to understand how each affects performance. Empirically, we showed how increasing state-action coverage often has a much greater impact on data efficiency than increasing reward density. Moreover, we demonstrated that the decreasing the augmented replay ratio can yield dramatic improvements to data efficiency. In fact, certain tasks are unsolvable unless the replay ratio is sufficiently small.

Our work has provided an initial study analyzing the benefits of DA. To better leverage DA, further work should study (1) how other properties of DAFs influence RL training, such as *relevancy* (Mitrano and Berenson, 2022), (2) how hyperparameters within a DA framework affect performance, and (3) how different RL algorithms affect performance. Our analysis focused on relatively low-dimensional sparse reward tasks with continuous actions, and findings may differ for tasks with dense rewards, discrete actions, or high-dimensional visual observations. Since our DA framework only integrates augmented data into model-free updates, it would be beneficial to extend this analysis to frameworks that use augmented data for auxiliary tasks such as representation learning rather than – or in addition to – policy optimization (Hansen and Wang, 2021; Raileanu et al., 2021).

## ACKNOWLEDGMENTS

Thanks to Brahma Pavse and to the anonymous reviewers for feedback that greatly improved our work. This research is supported in part by American Family Insurance through a research partnership with the University of Wisconsin-Madison's Data Science Institute and the Office of the Vice Chancellor for Research and Graduate Education at the University of Wisconsin — Madison with funding from the Wisconsin Alumni Research Foundation."

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

# Appendix

## Table of Contents

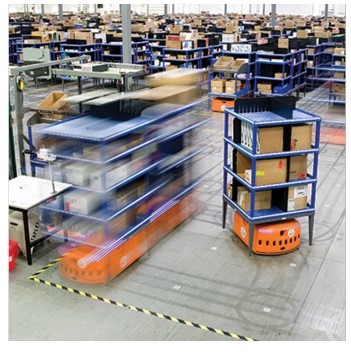 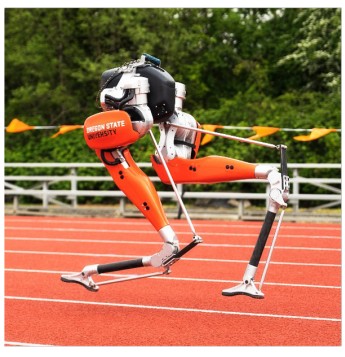 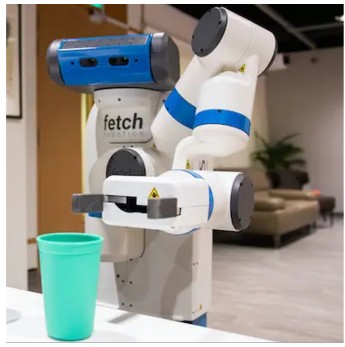

(a) Navigation     (b) Locomotion     (c) Manipulation

Figure 7: Common data augmentation functions. (7a, Amazon warehouse robots (2008)): If a robot is moving in free space, transition dynamics are often invariant to the agent's position. (7b, OSU's Cassie robot (2022)) Since robots are often symmetric about their sagittal axis, we can reflect the robot's left and right movements. (7c, Fetch robot (2023)) Objects move only if the agent contacts it. Thus, if the agent and object are not in contact, their dynamics are independent.

## A   DYNAMICS-INVARIANT DATA AUGMENTATION FUNCTIONS

In this section, we further motivate our focus on dynamics-invariant data augmentation functions. Specifying a dynamics-invariant data augmentation function requires knowledge of domain-specific invariances or symmetries. While domain knowledge may seem like a limitation, we observe in the literature and real-world RL applications that such invariances and symmetries are incredibly common and often require very little prior knowledge to specify. We provided a few examples:

1. Transition dynamics are often independent of the agent's goal state (Andrychowicz et al., 2017).

2. Objects often have independent dynamics if they are physically separated (Pitis et al., 2020; 2022), which implies that objects exhibit translational invariance conditioned on physical separation.

3. Several works focus on rotational symmetry of 3D scenes in robotics tasks (Wang et al., 2022; 2023), and many real-world robots are symmetric in design and thus have symmetries in their transition dynamics (Abdolhosseini et al., 2019; Mikhail Pavlov and Plis, 2018).

We include real-world tasks that exhibit one or more of these invariances in Fig. 7. We choose to focus on dynamics-invariant data augmentations because they have already appeared so widely in the literature. As RL becomes an increasingly widely used tool, we anticipate that domain experts will be able to identify new domain-specific augmentations and use them to further lower the data requirements of RL. These observations underscore the importance of identifying when and why different general properties of data augmentation will benefit RL.

While visual DAFs are also commonly used in RL, a study on such DAFs is beyond the scope of this analysis. Such studies will likely need to focus on different aspects of DA than the ones we considered in our work. Visual DAFs generate augmented data with the same semantic meaning as the original data, and this point has three implications:

1. It may be imprecise to say that visual DAFs increase coverage, because the observed and augmented data have the same semantic meaning.

2. Because the original and augmented observation have the same semantic meaning, the augmented reward will often be the same as the original reward. Thus, the concept of "reward density" does not apply to these augmentations.

3. Moreover, because these DAFs produce data which could never be observed through environment interaction (e.g. an agent would never receive a cropped, recolored, and rotated image from the environment), they primarily aid representation learning. In contrast, dynamics-invariant DAFs aid exploration.

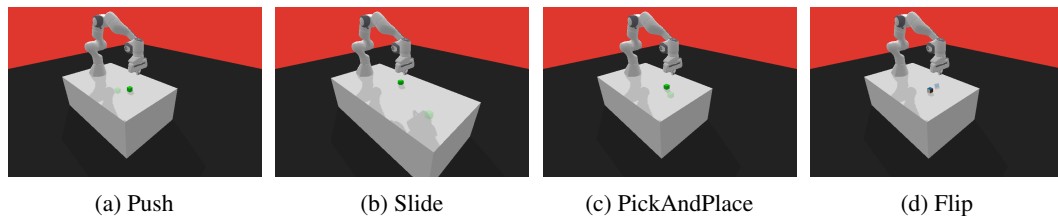

|  (a) Push  |  (b) Slide  |  (c) PickAndPlace  |  (d) Flip  |

Figure 8: Renderings of various Panda tasks. In PandaPush-v3 and Slide, the agent must move an object to a goal position. In PandaFlip-v3, the agent must rotate an object to a goal orientation.

## B PRIMARY ENVIRONMENTS FOR OUR EMPRICAL ANALYSIS

We use four tasks from `panda-gym` (Gallouédec et al., 2021) as the core environments in the main paper. Fig. 8 shows renderings for each task.

- **PandaPush-v3 (Push):** The robot must push an object to a goal location on the table. The goal and initial object positions are sampled uniformly at random from $(x, y) \in [-0.15, 0.15]^2, z = 0.02$.
- **PandaSlide-v3 (Slide):** The robot must slide a puck to a goal location on the table. The initial object position is sampled uniformly at random from $[-0.15, 0.15]^2$, while the goal $(x, y, z)$ is sampled from $x \in [0.25, 0.55], y \in [-0.15, 0.15], z = 0.015$.
- **PandaPickAndPlace-v3 (PickAndPlace):** The robot must pick up an object and move it to a goal location. With probability 0.3, the goal is on the table ($z = 0.02$), and with probability 0.7, the goal is in the air ($z \in (0.02, 0.2]$).
- **PandaFlip-v3 (Flip):** The robot must pick up an object and rotate it to a goal orientation. The initial object position is sampled uniformly at random from $[-0.15, 0.15]^2$, while the goal is a uniformly sampled orientation expressed a quaternion.

In Push, PickAndPlace, and Flip, the object is a cube with side length $0.04$. In Slide, the object is a cylindrical puck with height $0.03$ and radius $0.03$. The object's $z$ coordinate measures the distance between the center of the object and the table (*e.g.*, In Push, Slide, and PickAndPlace, $z = 0.02$ means the object is on the table).

Push, Slide, and PickAndPlace share a similar sparse reward structure. The agent receives a reward of 0 if the object is within 0.05 units of the goal and a reward of $-1$ otherwise. In Flip, the agent receives a reward of 0 if the object's orientation $\boldsymbol{q}$ is within 0.2 units from the goal orientation $\boldsymbol{q}_g$ under the following angle distance metric:

$$d(\boldsymbol{q}, \boldsymbol{q}_g) = 1 - (\boldsymbol{q} \cdot \boldsymbol{q}_g)^2 = \frac{1 - \cos(\theta)}{2}$$

where $\theta$ is the angle of rotation required to rotate $\boldsymbol{q}$ to $\boldsymbol{q}_g$. Otherwise, the agent receives a reward of $-1$.

In the toy 2D navigation task **Goal2D**, an agent must reach a fixed goal within 100 timesteps. The agent's state $(x, y, x_g, y_g)$ contains the coordinates of agent's positions $(x, y)$ and the goal's position $(x_g, y_g)$. At each timestep, the agent chooses an action $(r, \theta)$ and transitions to a new position:

$$\begin{aligned} x_{t+1} &= x_t + 0.05r\cos(\theta) \\ y_{t+1} &= y_t + 0.05r\sin(\theta) \end{aligned} \tag{1}$$

Thus, the agent moves at most 0.05 units in any direction. The goal position is fixed throughout an episode. The agent receives reward $+1$ when it is within 0.05 units of the goal and reward $-0.1$ otherwise. Agent and goal positions are initialized uniformly at random in $[-1, +1]^2$.

## C    DATA AUGMENTATION FUNCTIONS

In this section, we provide further details on the data augmentation functions introduced in Section 5.1.

- TRANSLATEGOAL: Goals are relabeled using a new goal sampled uniformly at random from the goal distribution. Reward signal is generated when the new goal is sufficiently close to the object's current position. To approximate the probability of this augmentation generating reward signal in each task, we sample 10M object and goal positions uniformly at random and report the empirical probability of the goal being sufficiently close to the object to generate reward signal.
  - **PandaPush-v3 (Push):** Reward signal is generated with probability approximately 0.075.
  - **PandaSlide-v3 (Slide):** The probability of generating reward signal depends on the current policy. The initial object and goal distributions are disjoint, so this augmentation can only generate reward signal if the agent pushes the object into the region $x \in [0.25, 0.55], y \in [-0.15, 0.15]$. If the object is in this region, this augmentation will generate reward signal with probability approximately 0.075.
  - **PandaPickAndPlace-v3 (PickAndPlace):** Reward signal is generated with probability approximately 0.04.
  - **PandaFlip-v3 (Flip):** Reward signal is generated with probability approximately 0.04.
- TRANSLATEGOALPROXIMAL($p$): Goals are relabeled using a new goal sampled from the goal distribution. With probability $p$, the new goal generates a reward signal, and with probability $1 - p$, no reward signal is generated. When generating an augmented sample with reward signal, the goal is set equal to the object's position plus a small amount of noise. and with probability $1 - p$, the object is moved to a random location sufficiently far from the goal that no reward signal is generated.

All Panda augmentation functions relabel the goal and reward. For Goal2D, we consider three data augmentation functions:

1. TRANSLATE: Translate the agent to a random position in $[-1, +1]^2$. This augmentation generates reward signal with probability approximately 0.019. We obtained this approximation by sampling 10M agent and goal positions uniformly at random and then computing the empirical probability of the goal being with 0.05 units of the agent.

2. ROTATE: Rotate both the agent and goal by $\theta \in \{\pi/2, \pi, 3\pi/2\}$. When sampling multiple augmentations of the sample observed transition, it is possible to sample duplicate augmentations.

3. TRANSLATEPROXIMAL($p$): Translate the agent to a random position in $[-1, +1]^2$. With probability $p$, agent's new position is within 0.05 units of the goal and generates reward signal, and with probability $1 - p$, agent's new position is more than 0.05 units from the goal and generates no reward signal.

All Goal2d augmentations modify the agent's state. TRANSLATE and TRANSLATEPROXIMAL($p$) modify the agent's position and reward but do not modify the goal. ROTATE affects the agent's position, the goal position, and agent's action, but does not change the reward.

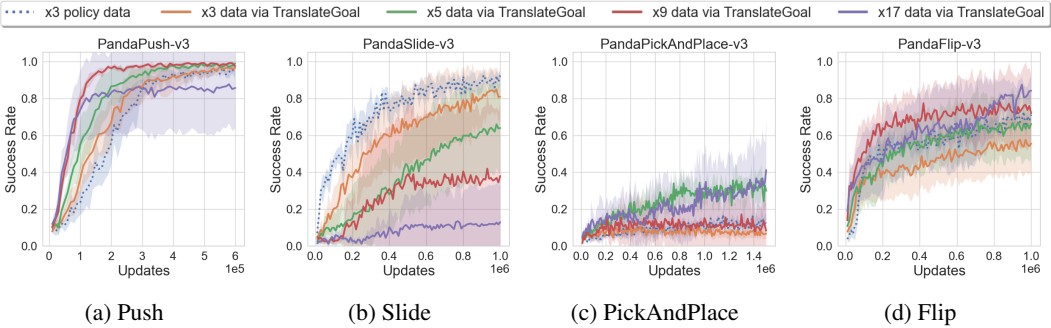

Figure 9: Increasing the update ratio while keeping the augmented replay ratio fixed at $\beta = 1$ may harm performance. We plot the mean over 10 seeds expect for agents that use very large batch sizes: "x9 data" and "x17 data" curves show 5 seeds; "x17 data" for PickAndPlace shows 3 seeds. Shaded regions are 95% confidence belts. These agents are trained using the same hyperparameters as those used in Fig. 6, though the hyperparameters are slightly different than those used in other figures. See Appendix G for further details.

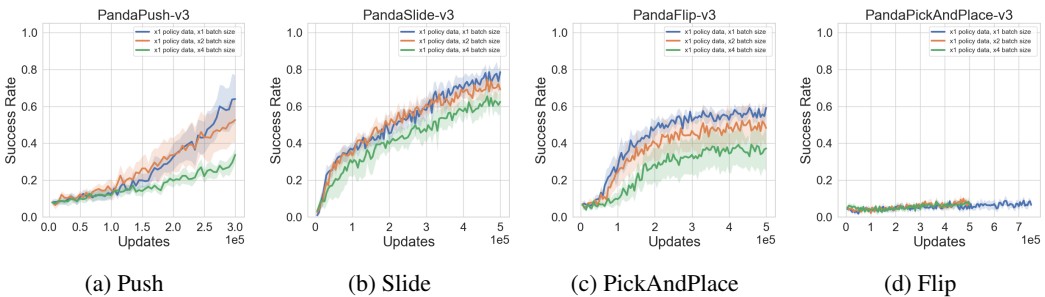

Figure 10: Increasing the batch size without increasing the amount of learning data available to the agent harms performance. We train agents using the hyperparameters listed in Table 2 with various batch sizes, *e.g.*, "x2 batch size" corresponds to learning with a batch size twice as large the batch size listed in Table 2.

# D ADDITIONAL EXPERIMENTS

In this appendix, we provide additional experiments supporting the core claims in the main paper.

## D.1 INCREASING THE UPDATE RATIO

In Section 5.1.4, we demonstrate that generating more augmented data to decrease the augmented replay ratio can drastically improve data efficiency. If we generated more augmented data by increasing the augmentation ratio, we could alternatively incorporate the additional augmented data by using more augmented data in each update (*i.e.*, increasing the update ratio). Fig. 9 shows agent performance as the augmentation ratio and update ratio increase proportionally. We additionally keep the replay age fixed by increasing the augmented buffer size proportionally. Learning is generally more data efficient with a larger update ratio, though it may harm performance as seen in Slide. Notably, the improvements in data efficiency from decreasing the replay ratio (Fig. 6) are similar or better than those produced from an increased update ratio and can be achieved at a much lower computational cost per update.

## D.2 INCREASING THE BATCH SIZE

In Fig. 3, agents with more training data available to them use larger batch sizes for updates, giving these agents a seemingly unfair advantage over agents that learn from less data. However, Fig. 10 shows that increasing the batch size without increasing the amount of data available to the agent

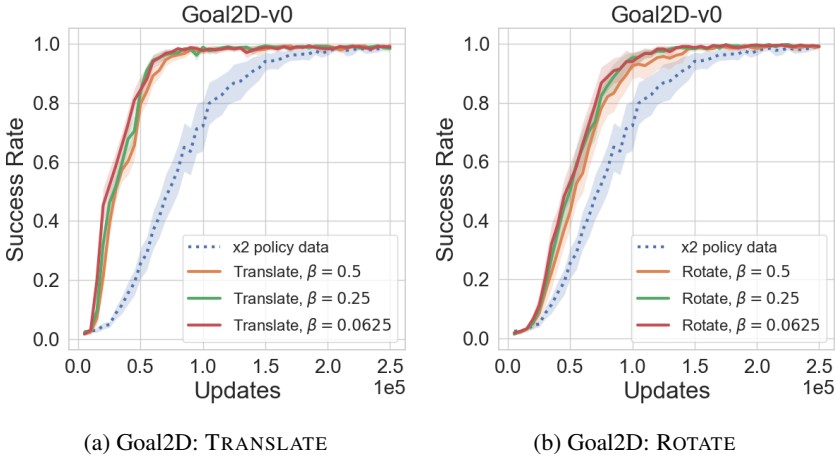

(a) Goal2D: TRANSLATE (b) Goal2D: ROTATE

Figure 11: (50 seeds) Decreasing the replay ratio while keeping the update ratio fixed at $\alpha = 1$ improves data efficiency.

harms performance, due to an increase in the expected number of times a transition is sampled for a gradient update (Fedus et al., 2020). By scaling the batch size with the amount of available learning data in Fig. 3, we keep the expected number of gradient updates per observed/augmented transition fixed across all experiments, providing a fairer comparison.

### D.3 GOAL2D AUGMENTED REPLAY RATIO

As in Section 5.1.4, we decrease the augmented replay ratio $\beta$ by generating more augmentations per observed transition. We scale the augmented replay buffer size proportionally and keep the ratio of augmented to observed data used in updates fixed at $\alpha = 1$. As shown in Fig. 11, a lower augmentation replay ratio increases data efficiency.

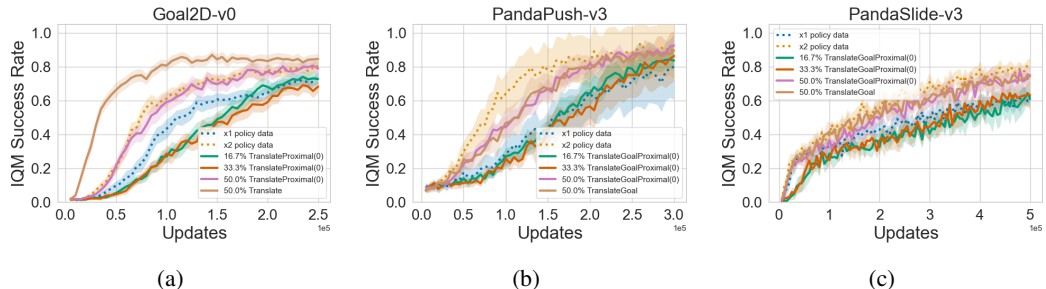

(a)  (b)  (c)

Figure 12: Learning with various mixtures of additional policy-generated data and TRANSLATEPROX-IMAL(0) or TRANSLATEGOALPROXIMAL(0) data. Each mixture doubles the agent's learning data. Solid lines denote averages over 10 seeds in Panda tasks and 50 seeds in Goal2D, and shaded regions denote 95% confidence intervals. The upper legend refers to Panda tasks results. We use an update ratio of $\alpha = 1$. Panda tasks use the same hyperapameters listed in Table 2.

## E  GENERALIZATION EXPERIMENTS

In this section, we investigate how state-action coverage, reward density, and the augmentated replay ratio affect an agent's generalization ability. For Push, Slide, and PickAndPlace, we train agents over one quadrant of the goal distribution and evaluate agents over the full distribution. For Flip, An agent that generalizes well will achieve a high success rate over the full goal distribution. In general, our observations regarding data efficiency in the main body of this work also apply to generalization.

### E.1  STATE-ACTION COVERAGE

State-action coverage results are shown in Fig. 12. An increase in state-action coverage via augmentation increases generalization. In the Panda tasks, using 50% TRANSLATEGOALPROXIMAL(0) data yields similar performance compared to increased using a 50% TRANSLATEGOAL data, indicating that coverage alone can largely explain the generalization improvements with TRANSLATEGOAL. In Goal2D, increased coverage yields better generalization, though a considerable gap nevertheless exists between TRANSLATEPROXIMAL(0) and TRANSLATE. Thus, reward density must play a larger role in Goal2D. We further investigate this point in the following section.

### E.2  REWARD DENSITY

Reward density results are shown in Fig. 13. In Goal2D, a relatively small increase in reward density dramatically improves generalization; TRANSLATEPROXIMAL(0) is roughly on-par with using twice as much policy-generated data, while TRANSLATEPROXIMAL(0.05) outperforms agents with x8 as much policy-generated data. In the Panda tasks, increasing reward density has little effect on data generalization. Just

### E.3  AUGMENTED REPLAY RATIO

Augmented replay ratio results are shown in Fig. 14. In Goal2D with TRANSLATE and both Panda tasks with TRANSLATEGOAL, reducing the augmented replay ratio $\beta$ improves generalization performance at convergence. ROTATE achieves 100% success for all values of $\beta$.

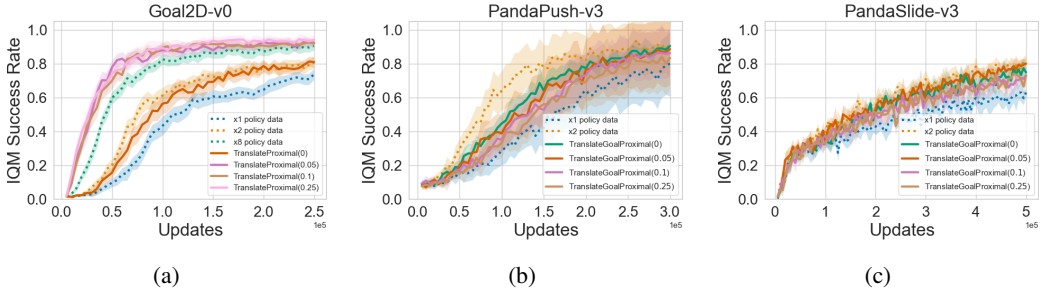

(a)  (b)  (c)

Figure 13: Learning with TRANSLATEGOALPROXIMAL($p$) and TRANSLATEPROXIMAL($p$) for various settings of $p$. Solid lines denote averages over 10 seeds in Panda tasks and 50 seeds in Goal2D, and shaded regions denote 95% confidence intervals. We use an update ratio of $\alpha = 1$. Panda tasks use the same hyperapameters listed in Table 2.

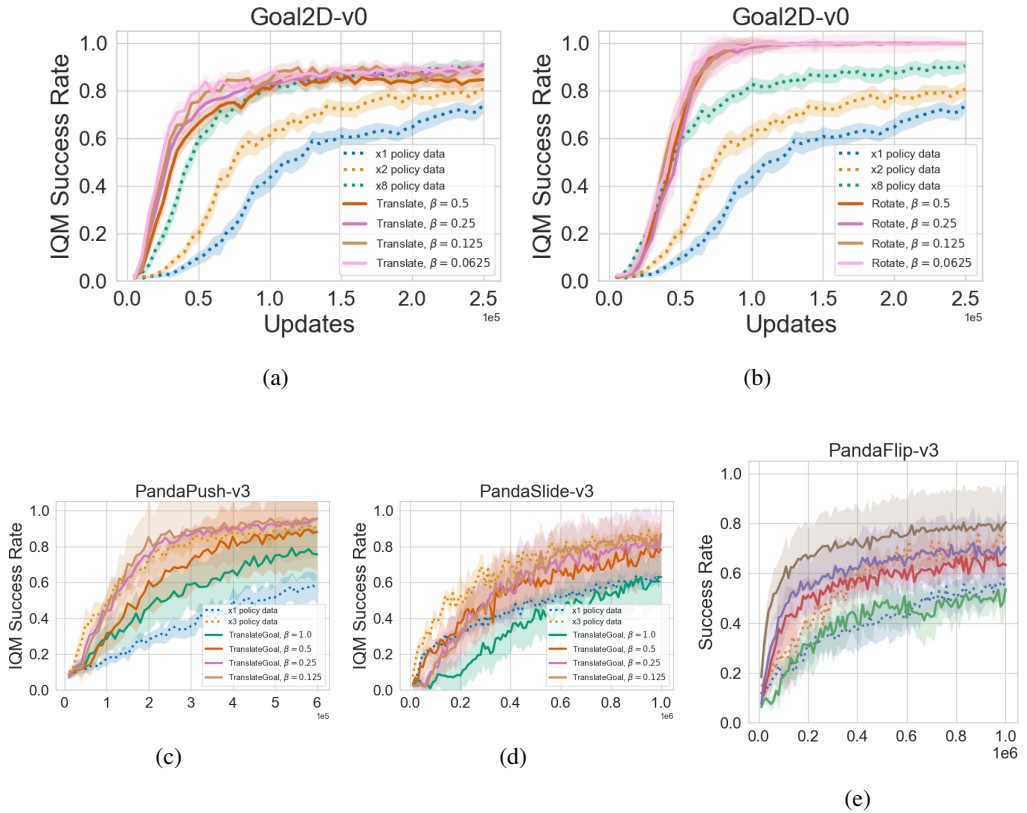

Figure 14: Decreasing the augmented replay ratio with various augmentations. Solid lines denote averages over 10 seeds in Panda tasks and 50 seeds in Goal2D, and shaded regions denote 95% confidence intervals. We use an update ratio of $\alpha = 1$ for Goal2D and $\alpha = 2$ for Panda tasks. Panda tasks use the same hyperapameters used to in Fig. 6.

| Environment | Modifications |
|---|---|
| InvertedPendulum, InvertedDoublePendulum | `cpole` `from_to` changed from $(0, 0, 0, 0.001, 0, 0.6)$ to $(0, 0, 0, 0, 0, 0.6)$. |
| Walker2d | `foot_left_geom` friction changed from 1.9 to 0.9 so that it matches the `foot_right_geom` friction. |
| Humanoid | `right_hip_y` armature changed from 0.0080 to 0.010 so that it matches `left_hip_y` armature. Specify `right_knee` stiffness of 1 so that it matches `left_knee` stiffness. Change `solver` from PGS to Newton. |

Table 1: Modifications to MuJoCo environments to ensure intuitive symmetries. We include modifications to InvertedPendulum tasks even though we do not consider them in our experiments. We leave them here as a reference for future work in data augmentation that may require symmetric InvertedPendulum dynamics.

# F  MuJoCo Experiments

In this appendix, we include additional experiments on the following dense reward, continuous state and action MuJoCo environments: **Swimmer-v4, Walker2d-v4, Ant-v4,** and **Humanoid-v4**. These experiments focus on the augmented replay ratio; the state-action coverage and reward density analysis provided in the main paper is tailored toward sparse reward tasks. With dense reward tasks, we may need to consider the full distribution of rewards in the replay buffer, not just the average.

## F.1  Environment Modifications

Some of the common MuJoCo environments do not exhibit symmetries that should exist intuitively (*e.g.* reflection symmetry, gait symmetry, etc.). We found two causes:

1. Asymmetric physics are explicitly hard-coded in the robot descriptor files. We believe these to be typos, and simply update values such that intuitive symmetries exist.

2. Symmetry-breaking numerical optimization algorithms are sometimes used to compute constraint forces and constrained accelerations. In particular, some environments use the Projected Gauss-Seidel (PGS) algorithm which performs sequential updates and therefore breaks symmetries where physics should be symmetric. To address this issue, we use Newton's method, which performs parallel updates and therefore preserves symmetric physics. We note that while Newton's method is MuJoCo's default algorithm, some robot descriptor files originally specify the use of PGS.

Table 1 describes all modifications made to environments to ensure intuitive symmetry. To simplify the creation of augmentation functions, we additionally exclude constraint forces and center-of-mass quantities from the agent's observations, since their interpretations are not well-documented and difficult to ascertain.

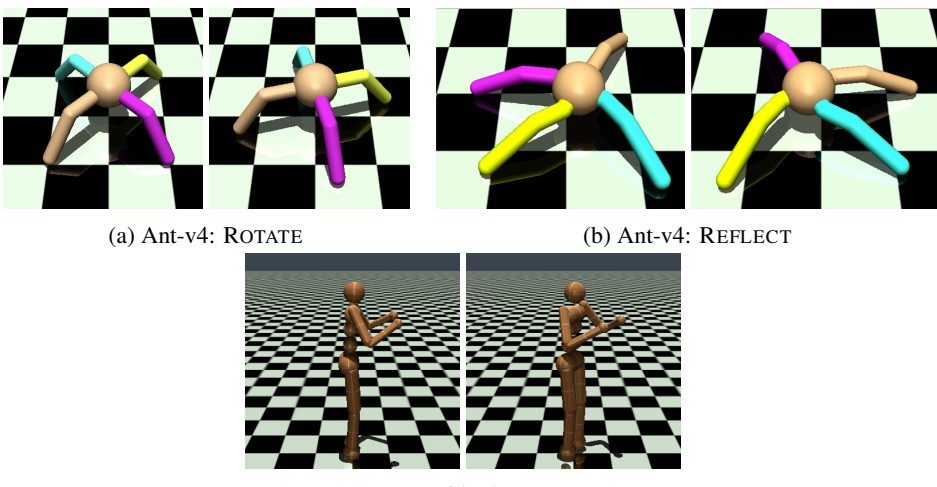

(a) Ant-v4: ROTATE                         (b) Ant-v4: REFLECT

(c) Humanoid-v4: ROTATE

Figure 15: Visualizations of some augmentations in MuJoCo environments.

## F.2 DATA AUGMENTATION FUNCTIONS

We consider the following dynamics-invariant augmentations:

- Swimmer-v4
    - REFLECT: Reflect joint angles and velocities about the agent's central axis. The reward is unchanged.
- Walker2d-v4
    - REFLECT: Swap the observations dimensions of the left and right legs. The reward is unchanged.
- Ant-v4
    - REFLECT: Swap the observations dimensions of the front and back legs. *Unlike the other reflections, this augmentation affects the reward.* In particular, if the observed transition moves forward with velocity $v$, the reflected transition will move backwards with velocity $-v$. Thus, this reflection flips the sign of the "forward progress" reward term in the reward function.
    - ROTATE: Rotate the agent's orientation by $\theta$ sampled uniformly at random from $[-\pi/6, \pi/6]$.
- Humanoid-v4
    - REFLECT: Swap the observations dimensions of the left and right arms/legs, and reflect torso joint angles, velocities, and orientation about the agent's central axis. The reward is unchanged.
    - ROTATE: Rotate the agent's orientation by $\theta$ sampled uniformly at random from $[-\pi/3, \pi/3]$.

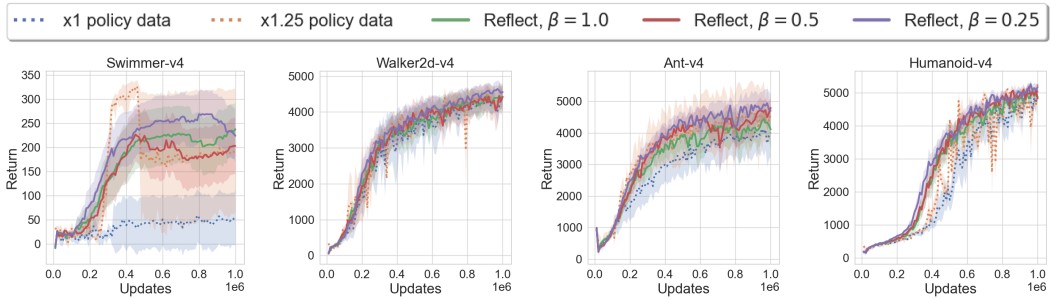

Figure 16: Lowering the augmented replay ratio for the REFLECT augmentation. Solid lines denote averages over 10 seeds, and shaded regions denote 95% confidence intervals.

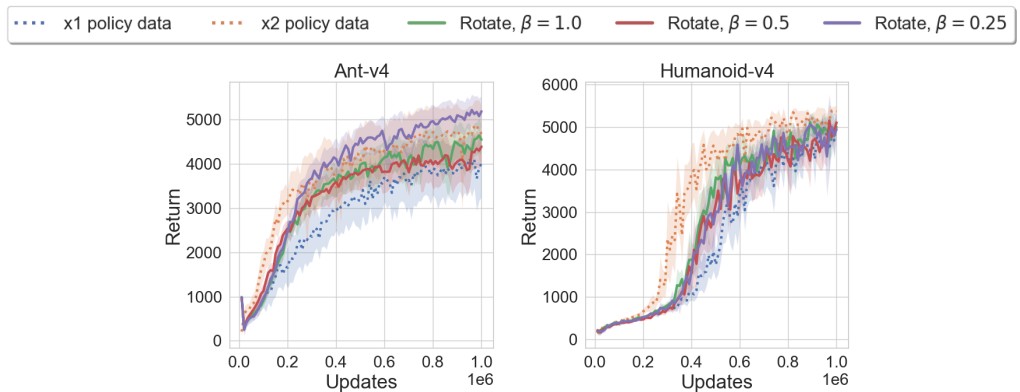

Figure 17: Lowering the augmented replay ratio for the TRANSLATE and ROTATE augmentations. Solid lines denote averages over 10 seeds, and shaded regions denote 95% confidence intervals.

### F.3 AUGMENTED REPLAY RATIO

We repeat the same augmented replay ratio experiments detailed in Section 5.1.4 for MuJoCo tasks. We decrease the replay ratio $\beta$ by generating more augmentations per observed transition while keeping the amount of augmented data used in policy/value function updates fixed, *i.e.*, we increase the augmentation ratio $m$ while fixing the update ratio $\alpha$. We consider all augmentations listed in Appendix F.2.

For the ROTATE augmentation, we fix the update ratio at $\alpha = 1$ and generate $m = 1, 2, 4$ augmented transitions per observed transition, corresponding to augmented replay ratios $\beta = 1, 0.5, 0.25$, respectively. Even though REFLECT can only generate a single augmented transition per observed transition, we can nevertheless study the augmented replay ratio as follows. Rather than generating an augmenting every observed transition, we instead augment transitions with some probability $p$ such that the augmented replay ratio is $1/p$ in expectation. We can think of $p$ as a fractional augmentation ratio. We fix the update ratio at $\alpha = 0.25$ and consider $p = 0.25, 0.5, 1$, again corresponding to augmented replay ratios $\beta = 1, 0.5, 0.25$, respectively.

Results are shown in Fig. 16 and 17. A lower augmented replay ratio with REFLECT yields slight improvements in data efficiency for all environments except Humanoid-b4, where performance is largely unchanged. We observe much larger improvements with a lower augmented replay ratio in our core sparse reward tasks (Section 5.1.4).

| | |
|---|---|
| Episode length | at most 50 timesteps |
| Evaluation frequency | 10,000 timesteps |
| Number of evaluation episodes | 80 |
| Number of environment interactions | 600K (Push), 1M (Slide, Flip), 1.5M (PickAndPlace) |
| Random action probability | 0.3 |
| Gaussian action noise scale | 0.2 |
| # of random actions before learning | 1000 |
| Observed replay buffer size (default) | $1 \cdot 10^6$ |
| Augmented replay buffer size (default) | $1 \cdot 10^6$ |
| Batch size (default) | 256 (Push, Slide), 512 (Flip, PickAndPlace) |
| Update frequency | Every 2 timesteps (observed replay ratio of 0.5) |
| Network | Multi-layer perceptron with hidden layers (256, 256, 256) |
| Optimizer | Adam (Kingma and Ba, 2014) |
| Learning rate | 0.001 |
| Polyak averaging coefficient ($\tau$) | 0.95 |

Table 2: Default hyperparameters used in all Panda tasks.

## G  TRAINING DETAILS

We use the Stable Baselines3 (Raffin et al., 2021) implementation of DDPG (Lillicrap et al., 2015) and TD3 (Fujimoto et al., 2018) with modifications to incorporate augmentation into the RL training loop. In Panda tasks, we use DDPG since we found that it performs substantially better than TD3. This observation was also made by  Gallouédec et al. (2021). All Panda experiments use the default hyperparameters presented in Table 2. These parameters are nearly identical to the those used by Plappert et al. (2018) and Gallouédec et al. (2021). The augmented replay ratio experiments in Fig. 6 and update ratio experiments in Fig. D.1 use different values for two hyperparameters specified below:

- Random action probability: 0
- Update frequency: Every timestep (observed replay ratio of 1)

We ran all experiments on a compute cluster using a mix of CPU-only and GPU jobs. This cluster contains a mix of Tesla P100-PCIE, GeForce RTX 2080 Ti, and A100-SXM4 GPUs. Due to limited GPU access, we only used GPUs for the augmented replay ratio experiments in Section 5.1.4, since these were our most computationally demanding experiments. We ran state-coverage, reward density, and generalization experiments on CPU only. CPU jobs took 12-36 hours each depending on the training budget, and GPU jobs took up to 16 hours each.

