# OpenReview forum: "Understanding when Dynamics-Invariant Data Augmentations Benefit Model-free Reinforcement Learning Updates"
_ICLR.cc/2024/Conference — ICLR 2024 poster_

### Official Review · Reviewer_Mi19 · 2023-11-01

**Soundness:** 4 excellent
**Presentation:** 4 excellent
**Contribution:** 3 good
**Rating:** 8
**Confidence:** 4

**Summary:**

This paper studies the effects of data augmentation in off-policy RL, notably the number of augmented transitions per observed transition, the number of augmentations per update, and the ratio of real data vs augmented data used in updates. The intermediate effects studied are the resulting diversity on state-actions and the diversity of rewards. They find that the state-action coverage is the most important outcome of data augmentation that leads to success, and this effect can provide equal (or even better) performance than drawing more real (diverse) data from the environment.

**Strengths:**

The paper is very well presented and clear: I wasn't able to find any flaws in the story nor the arguments. Motivation is very clear, and they do an excellent job isolating various factors in augmentation-based (off-policy) algorithms and provide clear experimentation. The experiments seem insightful and I believe that the conclusions are sound.

**Weaknesses:**

Very little weaknesses, though I think mostly I would like to see a little more clarity wrt ensuring f generates valid transitions.

**Questions:**

My primary question is wrt f, e.g., how is this guaranteed that f generates valid transitions, what are the consequences of f not generating valid transitions, what are the properties of f in the experiments provided, etc. Could you please clarify these points in the experimentation section as well as clarify earlier in the work whether or not f is chosen to have the characteristics noted (validity, etc). But how much does this matter? Is it ok to have some invalid transitions in the augmented data, how might these change the results, etc?

---

> ### Author Response · Authors · 2023-11-16
>
> We would like to thank the reviewer for their kind comments! We’re quite pleased to see that you found our work to be well-presented and that you appreciate the insights we draw from our empirical analysis.
>
> Below, we address your questions regarding how and why we generate “valid” augmented data. We would also like to point the reviewer to Appendix C in the supplemental material for a more in-depth description of all DAFs in our analysis.
>
> ## **Ensuring we generate valid augmented data:**
> Ensuring we generate valid augmented data requires domain knowledge; we know that the dynamics of the robot and objects in Panda tasks are independent of the goal state, and we know that translating and rotating the agent in Goal2D will generate data that respects the task’s dynamics. We want to note that while domain knowledge may seem like a limitation, we observe in the literature and real world RL applications that valid data augmentation functions are incredibly common and often require very little prior knowledge to specify. For example:
> 1. Transition dynamics are often independent of the agent’s goal state [1].
> 2. Objects often have independent dynamics if they are physically separated [2,3], which implies translational invariance conditioned on physical separation.
> 3. Several works focus on rotational symmetry of 3D scenes in robotics tasks [4,5], and many real-world robots are symmetric in design and thus have symmetries in their transition dynamics [6,7].
>
> We chose to focus on dynamics-invariant augmentations because they have already appeared so widely in the literature. Furthermore, as RL becomes an increasingly widely used tool, we anticipate that domain experts will be able to identify new domain-specific augmentations and use them to further lower the data requirements of RL. Hence, the importance of identifying when and why different general properties of data augmentation will benefit RL.
>
> ## **What if we generate non-valid augmented data?**
>
> Non-valid data, i.e. data that does disagrees with the task’s dynamics and/or reward function, can bias learning and reduce data efficiency (similar to how model uncertainty biases learning in model-based algorithms [8, 9]). Since our analysis aims to understand which aspects of DA contribute to observed improvements in data efficiency, our focus on dynamics-invariant DAFs removes a potential confounding factor in our analysis. An interesting future research question to consider might be “How much non-valid data can an agent tolerate without harming data efficiency?”
>
> Several prior works have used non-valid augmentations – especially those focusing on visual augmentations [*e.g.* 10, 11] – and we do think it is worth studying this class of DAFs. However, visual augmentations primarily aid representation learning, so such a study will likely need to focus on different aspects of DA than the ones we considered in our work and is thus beyond the scope of our analysis.
>
> Please let us know if our response clarifies your comments! If you have follow-up questions or comments, we are more than happy to discuss them with you.
>
> ## **References**
>
> [1] Adrychowicz et. al. "Hindsight Experience Replay." NeurIPS 2017.
>
> [2] Pitis et. al. Counterfactual Data Augmentation using Locally Factored Dynamics." NeurIPS 2020.
>
> [3] Pitis et. al. "MoCoDA: Model-based Counterfactual Data Augmentation."  NeuIPS 2022.
>
> [4] Wang et. al.  "On-Robot Learning with Equivariant Models." CoRL 2022.
>
> [5] Wang et. al. "The Surprising effectiveness of Equivariant Models in Domains with Latent Symmetry." ICLR 2023.
>
> [6] Pavlov et. al. "Run, Skeleton, Run: Skeletal Model in a Physics-Based Simulation." AAAI 2018.
>
> [7] Abdolhosseini et. al.  "On Learning Symmetric Locomotion." ACM SIGGRAPH 2019.
>
> [8] Moerland, Thomas M., et al. "Model-based reinforcement learning: A survey." Foundations and Trends in Machine Learning 16.1, 2023.
>
> [9] Polydoros and Nalpantidis. "Survey of model-based reinforcement learning: Applications on robotics." Journal of Intelligent & Robotic Systems 86.2, 2017.
>
> [10] Raileanu et al. “Automatic data augmentation for generalization in deep reinforcement learning.” arXiv:2006.12862, 2020.
>
> [11] Laskin et. al. "Reinforcement Learning with Augmented Data." NeurIPS 2020.

---

### Official Review · Reviewer_Y7KF · 2023-11-02

**Soundness:** 3 good
**Presentation:** 4 excellent
**Contribution:** 3 good
**Rating:** 8
**Confidence:** 3

**Summary:**

This paper is an empirical study of data augmentation in reinforcement learning, which has been shown to improve results. The authors hypothesize three possible causes for the improvement, 1) state-action coverage, 2) reward density, and 3) augmented replay ratio. For the off-policy and sparse-reward setting, a study is designed that disentangles these three causes as much as possible, on environments from panda-gym and 2D navigation. The results indicate that increasing state-action coverage and decreasing augmented replay ratio are more important, while increasing reward density is less important, although specific conclusions are task-dependent.

**Strengths:**

# Originality and significance #

As far as I know, this is the first systematic study on the underlying causes of the improvements from data augmentation in RL. As such, I think the results would be helpful for researchers designing similar algorithms in the future.

# Clarity #

The paper is written very clearly and the approaches are straightforward to understand. The separation of results into the main text and supplement is reasonable.

# Quality #

The approach used to disentangle the three hypothesized causes are clever. Uncertainties are also presented in the graphs.

**Weaknesses:**

The authors state many of the weaknesses of the work in the last paragraph of the paper: limited range of environments, restricted data augmentation framework, and incomplete set of properties investigated. One that they did not state was the limited range of algorithms. However, I think aside from the limited range of environments, those weaknesses are a reasonable consequence of the approach used to disentangle the three causes. Addressing them would make the disentanglement much harder.

**Questions:**

1. How were the algorithms used in the study chosen?
2. Is there intuition about how the differences in the tasks affects the results? For example, in figure 3 the "x2" and "x4" results for policy data and augmented data are similar, but this is not true in figure 1.

---

> ### Author Response · Authors · 2023-11-16
>
> First, we would like to thank the reviewer for their positive review. We’re pleased to see that you found our analysis to be well-written and clever, and our empirical findings to be useful to the research community. Below, we address your comments.
>
> # **Clarification on limitations:**
>
> Thank you for pointing out the additional limitation with respect to our RL algorithm choices – we’ll include it in our revisions.
>
> Different algorithms may be more or less capable of learning from augmented experience. We do think that it would be interesting to conduct another empirical study investigating how different algorithmic advances (e.g. multiple target networks [1], entropy regularization [2], n-step returns [3], etc.) may affect learning from augmented data. Several recent works have shown that n-step returns are critical to data efficient RL when learning with only observed data [4,5], and we hypothesize this may also be the case when learning with augmented data.
>
> # **How did we choose our algorithms?**
>
> We initially wanted to use TD3 throughout the paper; TD3 often performs much better than DDPG, and is much less expensive to run than SAC. However, we found that TD3 performed noticeably worse than DDPG on all Panda tasks. In fact,  Gallouedec et. al [6] (the creators of Panda robot tasks) made this same observation. Thus, we decided to use DDPG for Panda tasks and TD3 for Goal2D as well as all MuJoCo tasks in Appendix F.
>
> # **Why does data augmentation improve data efficiency more in Figure 1 than in Figure 3?**
>
> In the Goal2D task, we do see that additional augmented data improves data efficiency more than additional policy-generated data. In Panda tasks, augmented data is either just as good or slightly worse. We believe this difference arises because Goal2D is much simpler than the Panda tasks.
>
> The *absolute* improvement to data efficiency provided by DA is not a critical point in our analysis; we sought to understand how much different aspects of DA contribute to the observed *relative improvements* in data efficiency.
>
> Please let us know if we’ve addressed all of your comments. We are more than happy to discuss any follow-up questions or comments you might have!
>
> # **References**
> [1] Fujimoto et. al. "Addressing Function Approximation Error in Actor-Critic Methods." ICML 2018.
>
> [2] Haarnoja, Tuomas, et al. "Soft actor-critic: Off-policy maximum entropy deep reinforcement learning with a stochastic actor." ICML, 2018.
>
> [3] Sutton and Barto. "Reinforcement Learning: An Introduction." 2018.
>
> [4] Fedus et. al. “Revisiting fundamentals of experience replay.” ICML 2020.
>
> [5] Obando-Ceron et. al. "Small batch deep reinforcement learning." To appear in NeurIPS 2023.
>
> [6] Gallouedec et. al. “panda-gym: Open-Source Goal-Conditioned Environments for Robotic Learning” 4th Robot Learning Workshop: Self-Supervised and Lifelong Learning at NeurIPS 2021

---

> > ### Comment · Reviewer_Y7KF · 2023-11-22
> > **Thanks to the authors for your response**
> >
> > After reading the other reviews and responses, I have decided to keep my score. Although I agree with some of the other reviewers that the scope of the experiments is somewhat limited, I feel that the novelty and relevance of the work outweighs it and that the paper would be of interest to the community. I would encourage the authors to expand the variety of the experimental setup.

---

### Official Review · Reviewer_UZG5 · 2023-11-03

**Soundness:** 1 poor
**Presentation:** 2 fair
**Contribution:** 2 fair
**Rating:** 3
**Confidence:** 4

**Summary:**

The paper studies the effective use of the dynamics-invariant data augmentation (DA) in reinforcement learning (RL). The authors propose an empirical framework to investigate the effectiveness of DA in RL, where they particularly focus on the impact of the augmented replay ratio, governing the frequency of using augmentations. Based on this, they identify the ratio as one of critical hyperparameters in RL, and demonstrate that it is sometimes better to set the replay ratio lower, whereas, intuitively, it seems always better to use higher ratio as a part
of maximizing the benefit from the augmentation. Besides this, the authors provide a set of insights on using DA in RL, in particular, when reward is sparse.

**Strengths:**

The paper identifies a critical hyperparameter, the augmented relay ratio, in RL, which may overlooked before.

The authors propose a framework to empirically study the impact of data augmentation in RL.

The paper is well-written and easy to follow.

**Weaknesses:**

My major concerns are the weak messages of the empirical findings. The authors provide some useful (yet somewhat intuitive) insights on using data augmentation in RL, but no effective usages in practice. It would be better to clarify the use of their findings. In my understanding, the most useful message in practice is to consider tuning the augmented reply ratio. It would be helpful to provide practical strategies to use DA effectively, based on the empirical findings.

There has been a line of works to study the impact of data augmentation in RL, and maximize its efficacy in RL, e.g., [A,B,C]. It has been observed that maximally using the data augmentation in RL training is sometimes worse than not using the augmentation. In some sense, this coincides with this paper's major finding (low augmented replay ratio is better), although they focused on the vision-based RL, that is different from the one considered in this paper. It is necessary to discuss differences and coincidences between this work and existing ones.

In addition, the proposed framework to study the effectiveness of data augmentation seems straightforwardly designed. It would be helpful to clarify the technical novelty of the framework.

[A] Laskin, Misha, et al. "Reinforcement learning with augmented data." Advances in neural information processing systems 33 (2020): 19884-19895.
[B] Raileanu, Roberta, et al. "Automatic data augmentation for generalization in deep reinforcement learning." arXiv preprint arXiv:2006.12862 (2020).
[C] Ko, Byungchan, and Jungseul Ok. "Efficient Scheduling of Data Augmentation for Deep Reinforcement Learning." Advances in Neural Information Processing Systems 35 (2022): 33289-33301.

**Questions:**

It would be great if additional experiments, further discussion, or navigation to what I’ve missed can be provided to address the comments and questions in the weakness and what follows.

- Do the findings hold if DA is used for more than just augmenting data, e.g., policy distillation or representation learning? The other uses of DA may extract the full potential of DA in RL, and show somewhat different observations.

- As we all know, choosing hyperparameters is critical in any machine learning. Hence, I want to ensure that every hyperparameter has been optimized for each scenario.

---

> ### Author Response · Authors · 2023-11-16
> **Author Rebuttal (1/2)**
>
> We would first like to thank you for your review! We’re pleased that you found our work well written and noted the importance of our findings regarding the augmented replay ratio. We believe we can address your comments and questions with a few minor clarifications in our revisions, and we hope to discuss these clarifications with you.
>
> ## **Core messages of our empirical analysis:**
>
> In our submission, we provide two practical guidelines:
>
> 1. **Data augmentation (DA) should focus less on generating reward signal and more on increasing state-action coverage.** In our experiments, most – and sometimes all – of the benefits of a DAF can be explained by an increase in state-action coverage alone, and increasing reward density beyond a small threshold typically provides no further benefit. This guideline contrasts with the core motivation of HER and its follow-up works [1, 2] which focus on generating augmented reward signal to improve data efficiency.
> 2. **When it is possible to generate a large amount of augmented data, use this data to decrease the replay ratio** (rather than to increase the batch size used for updates.)
>
> We’re pleased that our second guideline was clear, and we hope our response clarifies the first guideline. In our revisions, we will better emphasize these guidelines.
>
> ## **Differences between our work and prior works studying visual augmentations:**
>
> This is a great question that we can clarify with a few minor additions to the related work section. There are a few fundamental differences between our work and the related works you’ve provided:
>
> 1. Most prior works – including all three you have listed – focused on introducing new types of data augmentation functions (DAFs) or frameworks and demonstrating that they can boost the data efficiency of RL. To the best of our knowledge, our work is the first to systematically investigate which aspects of DA are most responsible for observed improvements in data efficiency.
>
> 2. Raileanu et. al [3] and Ko & Ok [4] use augmented data for auxiliary tasks, while we focus on using augmented data for model-free  updates. Moreover, these auxiliary tasks aim to make the policy and value function invariant to augmentation, i.e. $\pi( \cdot | s) = \pi( \cdot | \tilde s)$ and $V(s) = V(\tilde s)$, and cannot be used with the dynamics-invariant DAFs we considered. For example, the optimal action generally changes if we translate the agent or goal, so it is undesirable to have $\pi( \cdot | s) = \pi( \cdot | \tilde s)$.
>
> 3. As you noted, all three works you listed consider visual augmentations that only transform the agent’s state and generate augmented data with the same semantic meaning as the original data. As such, they study DA for representation learning. In contrast, the dynamics-invariant DAFs we consider generate augmented data with a new semantic meaning and thus study DA to improve exploration.
>
> ## **Novelty of our framework:**
>
> The core novelty of our framework is that it permits fine-grained control over many hyperparameters relevant to DA, allowing us to systematically study different aspects of DA.
>
> Crucially, our framework guarantees that the agent uses the same ratio of augmented to observed data (or update ratio) in each update by sampling observed and augmented data from separate replay buffers. Existing DA frameworks [*e.g.*, 5,6] and applications of DA [*e.g.*, 7, 8] sample data from a shared replay buffer containing both augmented and observed data, so the update ratio can change across updates. Moreover, with a shared replay buffer, the update ratio and the augmentation ratio (the number of augmented samples generated per observed sample) are entangled; increasing the augmentation ratio increases the update ratio.
>
> Given that DA is so widely studied and that most prior work focuses on improving over baselines rather than understanding why certain methods perform well, we hope the analysis enabled by our framework is the first of many which move us towards the development of further practical DA guidelines.
>
> We hope our response clarifies our work's novelties with respect to the existing DA literature. If you have further questions, we are more than happy to discuss them!
>
> [1] Adrychowicz et. al. “Hindsight Experience Replay.” NeurIPS 2017
>
> [2] Li et. al. “Generalized hindsight for reinforcement learning.” NeurIPS 2020
>
> [3] Raileanu et al. “Automatic data augmentation for generalization in deep reinforcement learning.” arXiv:2006.12862, 2020.
>
> [4] Ko, Byungchan, and Jungseul Ok. "Efficient Scheduling of Data Augmentation for Deep Reinforcement Learning." NeurIPS, 2022.
>
> [5] Pitis et. al. “Counterfactual Data Augmentation using Locally Factored Dynamics.” NeurIPS 2020.
>
> [6] Adrychowicz et. al. “Hindsight Experience Replay.” NeurIPS 2017.
>
> [7] Fawzi et. al. “Discovering faster matrix multiplication algorithms with reinforcement learning.” Nature 2022.
>
> [8] Abdolhosseini et. al. “On Learning Symmetric Locomotion.” ACM SIGGRAPH 2019.

---

> > ### Comment · Reviewer_UZG5 · 2023-11-17
> > **A quick question**
> >
> > Thanks for the response. It will comment on the first response after some time to digest it. But, I want to ask if you can clarify the meaning and contribution of the "systematic" study on the data augmentation function. In my understanding, it is just an ablation study upon Algorithm 1 and a specific choice of network architecture, varying some hyperparameters. Please clarify technical contributions of the systemic study.

---

> > > ### Author Response · Authors · 2023-11-17
> > >
> > > Thank you for your response -- we appreciate the discussion!
> > >
> > > While one could view the augmented replay ratio experiments as an ablation study on Algorithm 1, the remaining experiments *keep all hyperparameters in Algorithm 1 fixed* and instead study how two general properties of different DAFs (state-action coverage and reward density) contribute to improvements in data efficiency. The DAFs we’ve chosen allow us to control state-coverage and reward density. By varying these factors, our study shows how these general properties impact RL with dynamics-invariant data augmentation and hence provides insight that can inform the future design of new data augmentation functions.
> > >
> > > In brief, the augmented replay ratio experiments study *how one should integrate augmented data into model-free updates*, while the state-action coverage and reward density experiments study *how one should design DAFs.*
> > >
> > > We're happy to provide further clarification if needed!

---

> ### Author Response · Authors · 2023-11-16
> **Author Rebuttal (2/2)**
>
> ## **Do the findings hold when DA is used for policy distillation or representation learning?**
>
> Our work focuses on understanding the benefit of integrating dynamics-invariant augmented data into model-free RL updates. Analogous studies focusing on policy distillation and representation learning are interesting directions for future work that we intend to investigate, but are beyond the scope of this analysis. We note that such studies will likely need to focus on different aspects of DA than the ones we considered in our work.
>
> In particular, prior DA works focusing on policy distillation [4] and representation learning [3] are designed to work with DAFs that generate augmented data with the same semantic meaning as the original data, such as visual augmentations. This point has three implications:
>
> 1. Because these augmentations produce data which could never be observed through environment interaction (e.g. an agent would never receive a cropped, recolored, and rotated image from the environment), they primarily aid representation learning rather than exploration.
> 2. It may be imprecise to say that these augmentations increase coverage, because the observed and augmented data have the same semantic meaning.
> 3. Because the original and augmented observation have the same semantic meaning, the augmented reward will often be the same as the original reward. Thus, the concept of  “reward density” does not apply to these augmentations.
>
> Having said this, the role of the augmented replay ratio can and should be studied for visual augmentation. Researchers have become increasingly interested in how the replay ratio affects learning [9-12], and lowering the replay ratio via augmentation may yield substantial improvements in data efficiency on visual tasks.
>
> ## **Hyperparameter tuning:**
>
> Appendix G in the supplemental material lists hyperparameters and training details for all experiments. For Panda tasks, we use the same hyperparameters used by Plappert et. al [13], the creators of the Fetch robot tasks analogous to the Panda robot tasks we consider in our work. Gallouedec et. al [14] (creators of Panda robot tasks) also use these hyperparameters. Plappert et. al found these hyperparameters by performing a sweep over many different hyperparameter combinations described in Appendix B of [13].
>
> We modified two hyperparameters in our replay ratio experiments (also noted in Appendix G). These experiments use a larger split of augmented data in each update, and we observed that these modifications improved performance in all tasks for both augmented and non-augmented agents.
>
> Again, please let us know if our response clarifies your comments. We would be happy to discuss any follow-up questions you may have!
>
> ## **References:**
> [3, from previous comment] Raileanu et al. “Automatic data augmentation for generalization in deep reinforcement learning.” arXiv:2006.12862, 2020.
>
> [4, from previous comment] Ko, Byungchan, and Jungseul Ok. "Efficient Scheduling of Data Augmentation for Deep Reinforcement Learning." NeurIPS, 2022.
>
> [9] Fedus et. al. “Revisiting fundamentals of experience replay.” ICML 2020.
>
> [10] Chen et. al. “Randomized Ensembled Double Q-Learning.” Arxiv, 2021.
>
> [11] Nikishin et. al. “The primacy bias in deep reinforcement learning.” ICML 2022.
>
> [12]  D'Oro et. al. “Sample-efficient reinforcement learning by breaking the replay ratio barrier.” ICLR 2023.
>
> [13] Plappert et. al. “Multi-Goal Reinforcement Learning: Challenging Robotics Environments and Request for Research.” Arxiv, 2018.
>
> [14] Gallouedec et. al. “panda-gym: Open-Source Goal-Conditioned Environments for Robotic Learning” 4th Robot Learning Workshop: Self-Supervised and Lifelong Learning at NeurIPS 2021

---

### Official Review · Reviewer_ynPP · 2023-11-08

**Soundness:** 2 fair
**Presentation:** 3 good
**Contribution:** 2 fair
**Rating:** 5
**Confidence:** 4

**Summary:**

This paper analyzes the effects of data augmentation in reinforcement learning. A new evaluation framework is built, and three important indexes of data-augmented RL are studied empirically.

**Strengths:**

The topic is interesting in providing a study on the effectiveness of data augmentation in reinforcement learning.

The paper presentation is clear, especially since the problem model and proposed framework are easy to follow.

**Weaknesses:**

There are many data augmentation ways, and the paper has also listed many of them. However, in experiments, only 2-3 basic ways are used. Considering the four typical testing environments, the conclusion of empirical analysis is quite limited.

The core idea of this paper can be very helpful for data-inefficient RL that needs augmentation, but it lacks theoretical analysis to support the proposed framework.

**Questions:**

Could you explain Definition 3 and its purpose for the following analysis? An example would be helpful.

---

> ### Author Response · Authors · 2023-11-16
>
> Thank you for your kind comments! We’re glad you found our work interesting, clear, and easy to follow. Below, we’ve addressed your comments and answered the question you posed. We hope to discuss these clarifications with you.
>
>
> ## **The purpose of Definition 3 (dynamics-invariant DAF):**
>
> We use definition 3 to define the class of data augmentation functions (DAFs) we focus on in this work. At a high-level, a dynamics-invariant DAF only generates augmented data that the agent could in principle observe through task interaction (i.e. data that agrees with the task’s dynamics and has the correct reward). Learning from augmented data that does not match the task’s dynamics can harm learning, similar to how model uncertainty biases learning in model-based algorithms [1, 2] . Since our analysis aims to understand how different aspects of DA affect the data efficiency of RL, our focus on dynamics-invariant DAFs eliminates a potential confounding factor.
>
> Upon consideration, we do not require the second half of this definition (“and if $f$ respects the stochasticity of $p$”) for our analysis and will remove it to prevent confusion. Definition 3 will become: “A DAF is dynamics-invariant if it is closed under valid transitions.” Intuitively, a DAF is dynamics-invariant if it takes valid transitions and produces new valid transitions.
>
> [1] Moerland, Thomas M., et al. "Model-based reinforcement learning: A survey." Foundations and Trends in Machine Learning 16.1, 2023.
>
> [2] Polydoros, Athanasios S., and Lazaros Nalpantidis. "Survey of model-based reinforcement learning: Applications on robotics." Journal of Intelligent & Robotic Systems 86.2, 2017.
>
> ## **Our choice of data augmentation functions (DAFs) in our experiments:**
>
> Thank you for this comment! Our decision to focus on a few DAFs comes from two practical considerations that we will clarify in the main paper.
>
> While our experiments focus on 3 dynamics-invariant DAFs – agent translation, agent rotation, and goal relabeling – these DAFs are extremely general and apply to many tasks. For instance, translation and rotation can be applied to most navigation tasks. In the Goal2D environment in Fig. 1, translation and rotation would still be valid DAFs if the agent were a more complex locomotor (e.g. a quadruped robot). Moreover, since task dynamics rarely depend on the goal state, goal relabeling can be applied to most goal-conditioned tasks.
>
> Other popular DAFs such as visual augmentations generate non-valid augmented data and are thus beyond the scope of our analysis. A study of visual augmentations would need to focus on different aspects of DA than the ones we considered in our work. For example, visual augmentations generally do not change a transition’s reward, so we cannot use them to study the effect of increasing reward density.
>
> We can emphasize both of these points in the paragraph just before section 5.1.1. We hope this clarifies our empirical design. Please let us know if you have further questions or comments; we would be more than happy to discuss them!
>
> ## **Regarding theoretical analysis:**
>
> We agree that theoretical analysis is important, however, like many prior DA works in RL, our work is empirical in nature. We can make suggestions for future theoretical analysis in the paper. Are there types of analysis that the reviewer thinks are particularly important for this line of work?

---

### Author Response · Authors · 2023-11-20
**Global Response**

We’d like to thank all reviewers for their thoughtful reviews! We’re especially pleased to see that three reviewers (UZG5, Y7KF, Mi19) recognized the significance of our conclusions and that all four reviewers found the paper well-written and easy to follow.

While prior works in data augmentation (DA) have demonstrated how different DA techniques improve data efficiency, to the best of our knowledge, our work is the first to study the degree to which different aspects of DA contribute to observed improvements in data efficiency. Towards this end, we introduce a general DA framework that enables us to study the three aspects of DA: state-action coverage, reward density (i.e. the fraction of reward signal in the agent’s learning data), and the augmented replay ratio (the number of augmented samples generated per environment interaction). We draw two core conclusions:

1. In our experiments, most – and sometimes *all* – of the benefits of a data augmentation function (DAF) can be explained by an increase in state-action coverage *alone*, and increasing reward density beyond a small threshold typically provides no further benefit. Thus, when designing new DAFs, **we advise practitioners to focus on generating augmented data with high coverage rather than generating additional reward signal.** These findings contrast with the core motivation of Hindsight Experience Replay (HER) [1] – one of the most widely applied DA techniques – which focuses on generating more reward signal.
2. Decreasing the augmented replay ratio greatly improves data efficiency. In fact, the Panda-PickAndPlace task is only solvable when the augmented replay ratio is sufficiently low. Thus, **when it is possible to generate large amounts of augmented data, we advise practitioners to use this data to decrease the augmented replay ratio** (rather than including more augmented data in each update).

Reviewers raised a few questions and comments that we believe are easily addressed by a few minor revisions to our submission:

* Review UZG5 asked for clarification on our first core conclusion and our framework’s novelty (conclusion 2 was clear).
* Reviewer ynPP mentioned that we only study 3 DAFs. We chose these DAFs because (1) they are *extremely general* and can be applied to many tasks, and (2) because they enable us to disentangle state-action coverage and reward density (unlike e.g. visual augmentations).
* Reviewers UZG5 and Mi19 asked questions regarding dynamics-invariant DAFs. We will be integrating our clarifications into our submission.

We will upload a revised submission soon. In the meantime, we'd like to invite the reviewers to discuss further comments or questions they may have; we will happily discuss them with you!

[1] Adrychowicz et. al. “Hindsight Experience Replay.” NeurIPS 2017

---

### Author Response · Authors · 2023-11-22
**Global Repose 2: Revisions**

Dear reviewers,

Thank you again for the helpful feedback! We've uploaded a new submission with a few minor revisions based on reviewer suggestions and comments. Revisions are shown in blue text:

* **Section 2:** Added how our work differs from prior work on visual augmentations.
* **Section 3:** Added clarifications on Definition 3 (dynamics-invariant DAF)
* **Section 5.1.3:** Clarified the importance of our state-coverage and reward density experiments.
* **Appendix A:** Further discussion on visual DAFs and how they're outside of the scope of our analysis.

We hope to engage in discussion with all reviewers; please let us know if our responses have clarified your comments! If there are any lingering questions, we're happy to discuss!

---

### Meta-Review · Area_Chair_MGRD · 2023-12-10

**Metareview:**

This paper is an empirical study of data augmentation in reinforcement learning, which has been shown to improve results. The authors hypothesize three possible causes for the improvement, 1) state-action coverage, 2) reward density, and 3) augmented replay ratio and study these in controlled empirical settings. The conclusion is perhaps not surprising: it seems like state-action coverage and augmented replay ratio are very important compared to reward density for effective performance.

Overall, the empirical analysis is presented in a convincing way. While the reviews were mixed, I think that most of the points raised by the reviewers were addressed. It would be nice to have some kind of a theoretical analysis in a linear setting (or something equivalent), but I don't think that's a major concern preventing accepting the paper (though certainly a lack of a formal analysis makes it a bit hard to go beyond a poster at this point).

**Justification For Why Not Higher Score:**

I think the paper makes some concrete analyses, and finds actionable conclusions, but there are still more avenues for rigorous analysis, including formal theoretical analysis.

**Justification For Why Not Lower Score:**

I think the paper would be of interest to the community and the analysis is done reasonably rigorously.

---

### Decision · Program_Chairs · 2024-01-16

Accept (poster)